# Genomic epidemiology of *Vibrio cholerae* during a mass vaccination campaign of displaced communities in Bangladesh

Alyce Taylor-Brown [1] ✉, Mokibul Hassan Afrad[2], Ashraful Islam Khan[2], Florent Lassalle [1], Md. Taufiqul Islam[2,3], Nabid Anjum Tanvir[2], Nicholas R. Thomson [1,4,5] ✉ & Firdausi Qadri[2,5] ✉

Ongoing diarrheal disease surveillance throughout Bangladesh over the last decade has revealed seasonal localised cholera outbreaks in Cox's Bazar, where both Bangladeshi Nationals and Forcibly Displaced Myanmar Nationals (FDMNs) reside in densely populated settlements. FDMNs were recently targeted for the largest cholera vaccination campaign in decades. We aimed to infer the epidemic risk of circulating *Vibrio cholerae* strains by determining if isolates linked to the ongoing global cholera pandemic ("7PET" lineage) were responsible for outbreaks in Cox's Bazar. We found two sublineages of 7PET in this setting during the study period; one with global distribution, and a second lineage restricted to Asia and the Middle East. These subclades were associated with different disease patterns that could be partially explained by genomic differences. Here we show that as the pandemic *V. cholerae* lineage circulates in this vulnerable population, without a vaccine intervention, the risk of an epidemic was very high.

The long-standing conflict in Myanmar's Rakhine State has seen hundreds of thousands of refugees belonging to the minority Rohingya Muslim (Forcibly Displaced Myanmar nationals; FDMNs) population seek asylum in Cox's Bazar, a coastal district of neighbouring Bangladesh. August 2017 saw the largest mass exodus of over 700,000 FDMNs moving into Cox's Bazar with the total population swelling to over a million people, with FDMNs now outnumbering the host population by a factor of two. This influx put a further strain on the already limited resources in the refugee camps. With high levels of childhood malnutrition, little or no access to sanitation and clean piped water, and little or no vaccine protection, this displaced population was extremely vulnerable to enteric diseases such as cholera[1].

Cholera is an acute watery diarrhoea caused by *Vibrio cholerae*, which historically has caused multiple pandemics. Our understanding of the genomic diversity of strains causing epidemic disease has shown that the majority of cases and outbreaks can be linked to a specific lineage of *V. cholerae*, referred to as "7PET" (seventh pandemic, El Tor biotype). Comparative genomic studies have shown that 7PET spread globally in successive but overlapping "waves"[2–4], linked to epidemics beginning in the 1960s and 1970s until the present day. By studying local patterns of disease, genomic data have also been used to differentiate 7PET *V. cholerae* from other *V. cholerae* lineages, which can be linked to important, but relatively low-level sporadic disease. These non-7PET lineages are referred to here as endemic lineages[2,5–8]. Importantly, both 7PET and endemic lineages can express the O1 type of the lipopolysaccharide (LPS)-based somatic (O) antigen, and carry the cholera toxin, the principal virulence determinant linked to epidemic disease[9]. Hence, in the absence of genomic data, it can be difficult to differentiate endemic lineages from epidemic sublineages of 7PET based on routine phenotyping alone, especially during low incidence periods/settings.

[1]Parasites & Microbes Programme, Wellcome Sanger Institute, Wellcome Genome Campus, Hinxton, Cambridge CB10 1SA, UK. [2]Infectious Diseases Division, International Centre for Diarrhoeal Disease Research, Bangladesh (icddr,b), Dhaka, Bangladesh. [3]School of Medical Science, Griffith University, Gold Coast, QLD, Australia. [4]London School of Hygiene and Tropical Medicine, London WC1E 7HT, UK. [5]These authors jointly supervised this work: Nicholas R. Thomson, Firdausi Qadri. ✉e-mail: at21@sanger.ac.uk; nrt@sanger.ac.uk; fqadri@icddrb.org

Cholera is endemic to Bangladesh, with an estimated 66 million people at risk (~40% of the population) and a case fatality rate of 3%[10]. To address this, long term sentinel surveillance was instituted in 2014 by the International Centre for Diarrhoeal Disease Research, Bangladesh (icddr,b), and the Institute of Epidemiology, Disease Control and Research (IEDCR, Government of Bangladesh) to survey and direct cholera control efforts in this region[11,12]. Following detection of serogroup O1 *V. cholerae* in Cox's Bazar, and with ongoing influx of FDMNs, the Government of Bangladesh successfully lobbied to access the global stockpile of World Health Organisation (WHO) pre-qualified whole-cell killed bivalent (O1 and O139) Oral Cholera Vaccine (OCV), Shanchol™[1]. This stockpile is limited and thus heavily controlled, with provisions reserved for epidemic response rather than in endemic settings[13]. The feasibility of emergency deployment of the OCV from the emergency stockpile was previously demonstrated in areas affected by civil conflict and natural disaster, with variable levels of effectiveness, such as South Sudan[14], Yemen[15], and areas without a history of cholera such as Haiti[16–18], where a lack of sanitation and public health facilities resulted in well-documented large-scale cholera epidemics with high mortality and morbidity. The size of these campaigns ranged from ~100,000[19] to ~920,000[15] total doses, split between the first and second dose. OCV has also been shown to lower the risk of severely dehydrating cholera in urban endemic settings[20,21], with protective efficacy shown for up to five years[22]. In October 2017, over 900,000 doses of OCV Shanchol™ were given as a single or first dose to FDMNs aged over 1 year old, with those aged 1–4 years receiving a second dose[1]. The second phase saw an additional ~879,000 FDMNs and the adjacent host population vaccinated in May 2018.

Despite detecting serogroup O1 *V. cholerae* in the host Bangladeshi (BGDN) communities, it was unknown whether the aetiological agents were low-risk, endemic or high-risk, 7PET strains. Here, we sought to understand the nature of the lineages circulating in Cox's Bazar, linking this to the dynamics of cholera in the Bay of Bengal region, as well as globally. We sequenced *V. cholerae* isolates obtained from stool samples collected between July 2014 and November 2019 from BGDNs and FDMNs presenting to sentinel surveillance sites with acute watery diarrhoea (AWD). Phylogenomic, spatiotemporal and epidemiological analysis revealed multiple lineages with contrasting phylodynamics within the high-risk epidemic lineage, 7PET.

## Results

### Cholera prevalence and epidemiology prior to and following a mass vaccination campaign in Cox's Bazar, Bangladesh

Sentinel surveillance for AWD in Cox's Bazar initially targeted the host BGDN communities and used the case definition outlined by the WHO. This definition is fully defined in methods, but in brief, an AWD case was defined as: "any patient who has had three or more loose or watery stools in the past 24 h, or three or fewer loose/liquid stools causing dehydration". Surveillance was carried out across all age groups, with four patients that met the case definition enrolled per day, five days a week, with the presence of *V. cholerae* confirmed by culture upon receipt of the sample at icddr,b (see methods and ref. 12).

Surveillance data collected between 2014 and 2018 showed a cholera prevalence of 8.4% from all AWD cases tested per year in the BGDN host population residing in Cox's Bazar, adjacent to the refugee camps[12]. Surveillance was subsequently extended to include FDMNs and BGDNs residing in the refugee camps[12], where there was little information on cholera incidence prior to the mass vaccination campaign. Vaccination began in October 2017, first targeting the FDMN subpopulation, and later the adjacent BGDN host population[1].

We sequenced isolates obtained from 150 BGDNs residing outside of, but adjacent to the refugee camps (host population; HP_BGDN) and from 42 BGDNs and 31 FDMNs residing in the refugee camps (RC_BGDN and RC_FDMN, respectively; Table 1). The sample sets spanned before, during, and after the mass vaccination periods, with host population BGDN samples collected from July 2014 until March 2019, and the RC samples collected over a shorter but more recent timeframe, September 2017 to November 2019. Samples were obtained from *V. cholerae*-positive AWD patients attending ten surveillance sites throughout the Cox's Bazar Sadar, Ukhia, and Teknaf upazilas (administrative divisions; Table 1). The majority of our *V. cholerae*-positive samples from FDMNs (*n* = 21/31) were obtained from Kutupalang (in the Ukhia upazila), the largest RC in the world, while most of the BGDN samples from refugee camps originated in Teknaf union (small settlement), within the Teknaf upazila (*n* = 26/42; Table 1 and Fig. 1). Most samples from BGDNs living outside of the refugee camps (140/150) were from those living in the adjacent Cox's Bazar Sadar upazila.

The majority of the patients included in this study were adults over 18 years old (*n* = 135); 25 patients were aged from 5 to 18, and 62

**Table 1 | Stool samples collected for whole genome sequencing of *V. cholerae* isolates**

| Upazila[a] | Location | Surveillance site | Number of samples per subpopulation[b] | | | Total number of samples | |
|---|---|---|---|---|---|---|---|
| | | | HP_BGDN | RC_BGDN | RC_FDMN | Per site | Per location |
| Cox's Bazar Sadar | | | 150 | | | 150 | 150 |
| Ukhia | Balukhali RC | Balukhali PHC | | | 2 (1) | 2 | 3 |
| | | IOM Balukhali PHCC | | | 1 (1) | 1 | |
| | Burmapara SS | IOM Burmapara | | | 1 | 1 | 1 |
| | Kutupalang RC | IOM Kutupalang D4 | | | 4 (3) | 4 | 26 |
| | | Kutupalang Ganshashtho | | | 3 (2) | 3 | |
| | | Kutupalang MSF | | 5 (1) | 13 (7) | 18 | |
| | | Kutupalang UNHCR | | | 1 | 1 | |
| | Raja Palong union[c] | Ukhia Health Complex | | 11 | 1 | 12 | 12 |
| Teknaf | Nayapara RC | Nayapara UNHCR | | | 3 (2) | 3 | 3 |
| | Teknaf union | Teknaf Health Complex | | 26 | 2 (1) | 28 | 28 |
| Total | | | 150 | 42 | 31 | 223 | |

[a]Upazilas are administrative divisions.
[b]Numbers in parentheses represent the numbers of vaccinated patients.
[c]Unions are small settlements.
*RC* Refugee camp, *SS* Surveillance site, *HP_BGDN* Bangladeshi National residing adjacent to RCs, *RC_BGDN* Refugee camp Bangladeshi National, *RC_FDMN* Refugee camp Forcibly Displaced Myanmar National, *IOM* International Organisation for Migration, *MSF* Médecins Sans Frontières, *UNHCR* United Nations High Commissioner for Refugees, *PHC* Primary Health Care, *PHCC* Primary Health Care Centre.

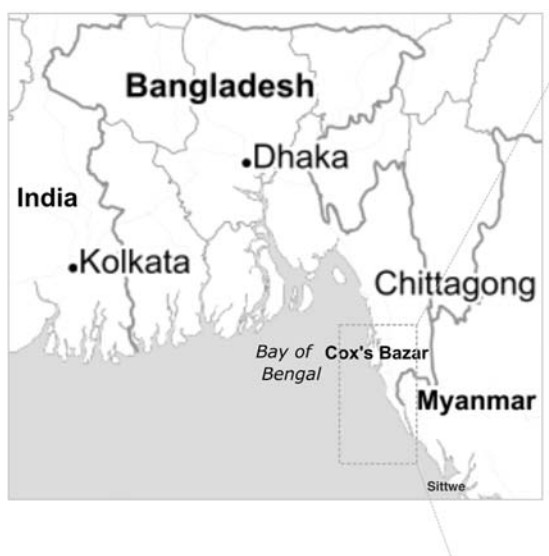

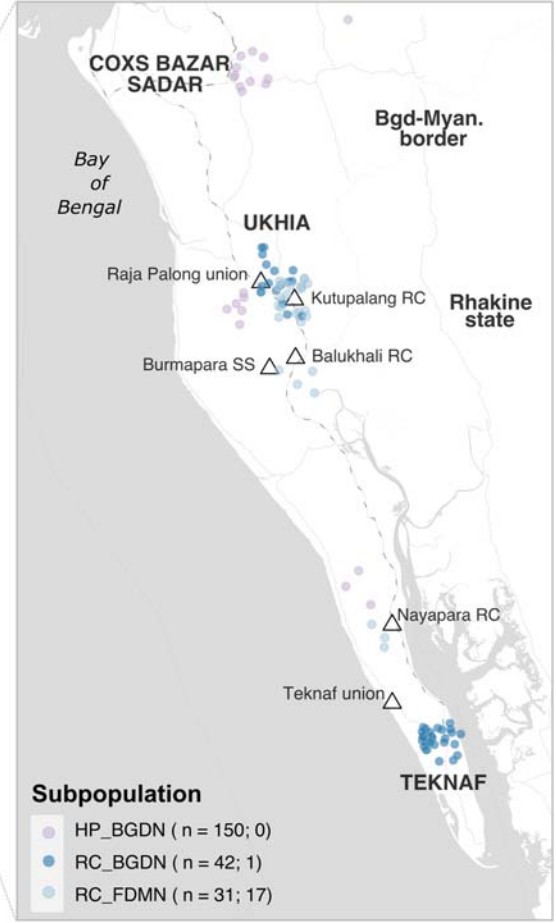

**Subpopulation**
- HP_BGDN ( n = 150; 0)
- RC_BGDN ( n = 42; 1)
- RC_FDMN ( n = 31; 17)

**Fig. 1 | Geographical distribution and context of samples originating from Cox's Bazar, Bangladesh, for this study.** Geographical context of Cox's Bazar (boxed area) within the Bay of Bengal, including India, Bangladesh and Myanmar. The boxed region is blown up to show the geographical distribution of the isolates sequenced in this study. Points are coloured by subpopulation (see key), with sample numbers shown in the key, followed by the number of vaccinated patients per group. Upazilas are marked using capital letters while refugee camps are written in lowercase and denoted by triangles. For the HP_BGDN subpopulation, samples are plotted using GPS location data collected at the time of sampling, while the RC (BGDN and FDMN) samples are plotted using latitude and longitude information for the surveillance site. Maps were obtained via StamenMaps from Stamen Design (www.maps.stamen.com/toner), under a Creative Commons Attribution (CC BY 3.0) license (https://creativecommons.org/licenses/by/3.0/) and plotted using the R package ggmap, with excess point overlap avoided using geom_jitter. SS Surveillance site, RC refugee camp.

were under 5 years old (Supplementary Data S1). The number of female and male patients were 118 and 105, respectively. Eighteen samples included here were from patients who developed symptoms after being vaccinated. Of these, 12 were females between two and 57 years old and six were males under five years old. All were from FDMNs residing in refugee camps in Balukhali, Kutupalang, Nayapara and Teknaf, except for one who was a BGDN residing in the Kutupalang RC. The vaccination status of 10 patients included in this study (six RC_BGDNs and four FDMNs) was unknown, and no BGDNs residing in the adjacent host population were known to be vaccinated prior to symptom onset (Table 1).

### Vibrio cholerae isolates from Cox's Bazar belong to multiple distinct clades of 7PET

To classify the *V. cholerae* circulating in Cox's Bazar, we constructed a phylogeny of *V. cholerae* isolates based on single nucleotide polymorphisms (SNPs; n = 490,634) in a concatenated core gene alignment (n = 2834 genes). To our 223 genomes sequenced in this study, we added 144 *V. cholerae* genomes selected to represent the diversity within the species (Supplementary Data S2). All isolates sequenced in this study fell within the 7PET lineage, clustering near *V. cholerae* reference genome N16961 (Supplementary Fig. S1).

Next, by combining our 223 newly sequenced genomes with a collection of 2384 previously published 7PET genomes (Supplementary Data S2), we constructed a reference-mapped whole genome SNP phylogeny (inferred from 15,420 variable sites; Supplementary Fig. S2). Hierarchical Bayesian Analysis of Population Structure (hierBAPS; see methods), showed that of the eight resultant high level 7PET hierBAPS Clades, all but one of the genomes from Cox's Bazar sat alongside genomes from South Asia within two distinct, but monophyletic clades of pandemic Wave 3 of 7PET: Clade 5 (n = 171) and Clade 3 (n = 51; Fig. 2a). The phylogeny was further divided to a lower level, which showed that our genomes belonged to two clade 5 subclades (5.19 and 5.20) and a single Clade 3 subclade (3.9) (Fig. 2a), which were present in different combinations throughout the Bay of Bengal (Fig. 2b) across overlapping time periods (Fig. 2c). Unusually, the remaining isolate genome from this study fell within a pandemic Wave 1 sublineage of 7PET (Supplementary Fig. S2).

Newly sequenced genomes belonging the two Clade 5 subclades grouped with contemporaneous Dhaka and Kolkata isolates (Fig. 2, Fig. 3) with, overall, Clade 5 being comprised of genomes from isolates collected between 1992 and 2018 predominantly from South Asia (Fig. 2a, Supplementary Figs. S3, S4). Clade 5 corresponds to a sublineage encompassing previously described lineages: "3.B"[5], "Asian

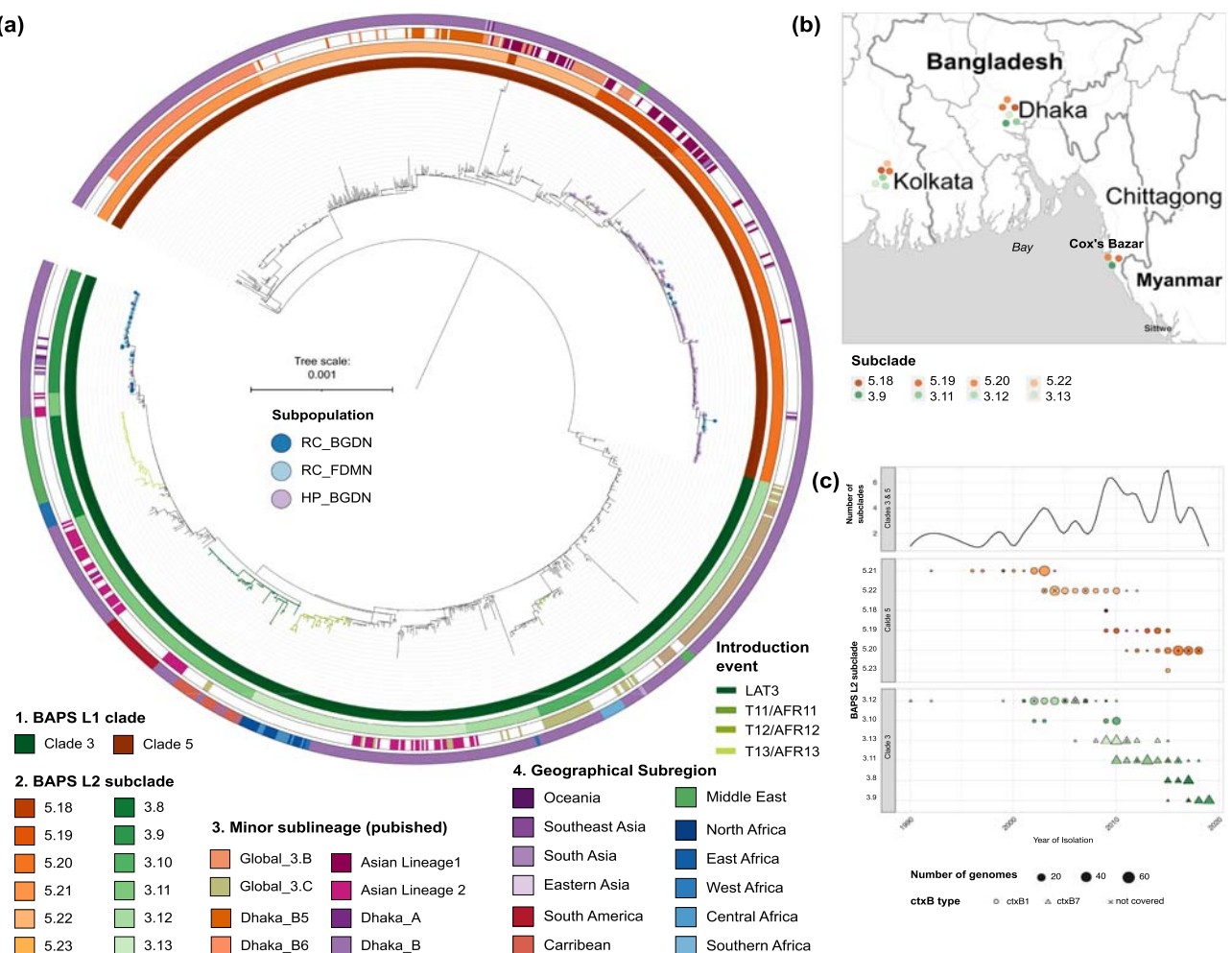

**Fig. 2 | Phylogenetic and spatiotemporal distribution of *Vibrio cholerae* 7PET Clades 3 and 5. a** Subtree of the whole genome SNP maximum-likelihood out-group-rooted phylogenetic tree of 223 *V. cholerae* strains from Cox's Bazar from this study and 872 published genomes from Clades 3 and 5 (see Supplementary Fig. S2 for full 7PET tree). Tree tips are coloured by subpopulation (see key). Coloured strips denote hierBAPS Clade (level 1 clustering) and subclade (level 2 clustering), previously published sublineages, and geographical subregion. Introduction events/sublineages are denoted by coloured branches (see key). **b** Subclade distribution across the cities in the Bay of Bengal (Bangladesh and India). Maps were obtained via StamenMaps from Stamen Design (www.maps.stamen.com/toner), under a Creative Commons Attribution (CC BY 3.0) license (https://creativecommons.org/licenses/by/3.0/) and plotted using the R package ggmap. **c** Temporal distribution of Clade 3 and 5 subclades. Genome counts are denoted by the size of the point, while the shape denotes the *ctxB* type (see key). Above the dot plot is a line graph showing the number of Clade 3 and 5 subclades present per year.

Lineage 1"[23,24] and "Clade B"[25]. Fourteen subclade 5.19 *V. cholerae* genomes originating from the host BGDN communities, collected between July 2014 and July 2015, sat alongside isolates from Pakistan (*n* = 7), Iran (*n* = 4), India (*n* = 1) and Bangladesh (*n* = 23)[26] (Fig. 2a, Fig. 3a). This subclade was not detected in the refugee camps. Subclade 5.20 is so far restricted to Bangladesh, with isolates from Cox's Bazar (*n* = 132, 12 and 13 BGDN, RC_BGDN and RC_FDMN isolates, respectively) interspersed with isolates from Dhaka (*n* = 30; Figs. 2a, 3b). Isolate genomes in this subclade were collected from patients residing in nine out of the ten surveillance sites represented in this study. Another Clade 5 subclade, 5.18, also appears to be restricted to India and Bangladesh[23,24] (Fig. 2a, Supplementary Figs. S3, S4). Subclade 5.21 and 5.22 comprise samples from South, East and Southeast Asia[4,8,23,27], spanning years preceding this study; and subclade 5.23 is so far restricted to the Middle East[8] (Fig. 2a, Supplementary Figs. S3, S4).

All Clade 3 genomes sequenced here belonged to subclade 3.9; 47/51 were derived from isolates collected in the refugee camps following vaccination, with the remaining four being from BGDN patients (Fig. 2a & 3c). Clade 3 corresponds to the previously described sublineage "3.C", "Asian Lineage 2" and "Clade A"[5,23–25]. Subclade 3.9 so far appears restricted to Bangladesh and India with the earliest genome

seen in Dhaka in 2015. This contrasts with Clade 3 sublineages which have radiated globally: first seen in the Bay of Bengal in 1990 (Figs. 2a, c & Supplementary Fig. S4). Clade 3 lineage isolates have been detected between 2010 and 2018, in the Middle East[8], South America[6] ("LAT-3"; subclade 3.11) and Haiti[28] (subclade 3.11), as well as being linked to multiple cholera introductions into Africa[8,29] ("T11/AFR11" - "T13/AFR13"; subclades 3.10, 3.13, 3.8). It is important to note that this clade was responsible for outbreaks linked to humanitarian crises in Yemen and Haiti (Fig. 2a, Supplementary Fig. S3), where it has recently re-emerged after being declared cholera-free[30,31]. Globally, the emergence and spread of Clade 3 overlapped in time and space with several Clade 5 subclades, with multiple different subclades of each circulating concurrently in any one year (Fig. 2c and Supplementary Fig. S4).

As mentioned above, one genome (RK2170112) belonged to 7PET pandemic Wave 1 (Clade 4; Supp Fig. 2). This isolate was collected in October 2017 from a 15-month-old FDMN residing in Kutupalang RC (vaccination status unknown). Based on the core gene alignment, RK2170112 was identical to two Bangladeshi genomes collected in 1979 (Inaba strain A22; accession number ERR025386) and 2003 (strain CIRS239; accession number ERR425233), the latter of which was the sole genome in the previously described Dhaka lineage "B2". Unlike these

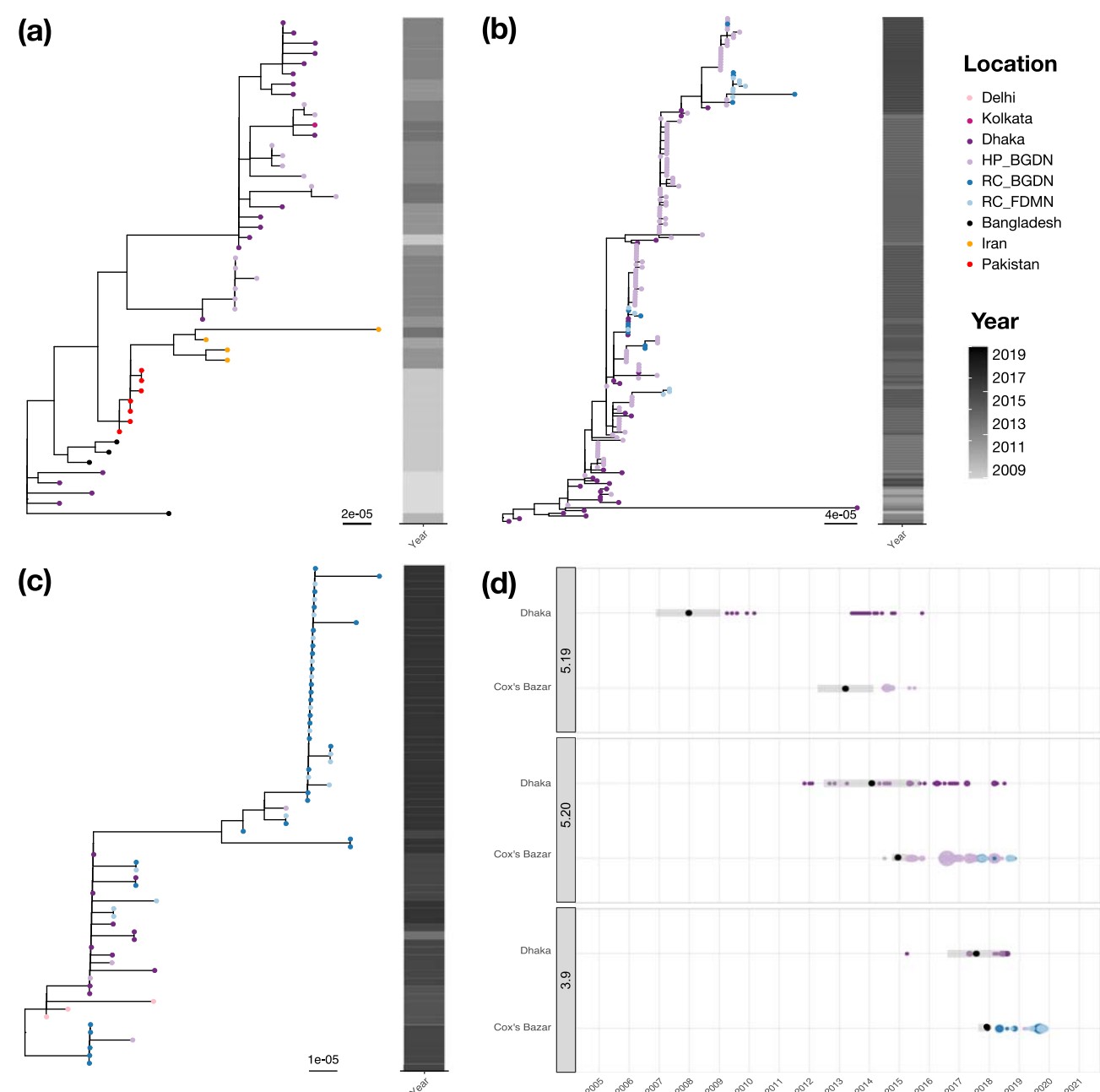

**Fig. 3 | Phylogenetic and spatiotemporal dynamics of *Vibrio cholerae* in Cox's Bazar, Bangladesh. a–c** Whole genome SNP maximum-likelihood outgroup-rooted phylogenetic subtrees of subclades 5.19 (**a**), 5.20 (**b**) and 3.9 (**c**). Tip colours denote geographical location or subpopulations from which our samples were isolated (see keys). Coloured strip denotes year of sampling (see key). Scale bars represent substitutions per site. **d** Temporal distribution of subclades present in Cox's Bazar and Dhaka. Genome counts are denoted by the size of the point. Points are coloured by geographical location or subpopulations as in (**a–c**). Black points denote the estimated tMRCA, with 2× standard error shown by the grey shading.

other Wave 1 genomes, which carry a *ctx*B3 allele, RK2170112 carries a ~7000 bp deletion between *rst*C and *rst*B2 that removed the RS1 and most of the CTX prophage, including the *ctx* locus that encodes both cholera toxin subunits. RK2170112 does carry an intact copy of the *Vibrio* pathogenicity island (VPI) encoding the *tcp*A gene – the receptor for CTX – and so likely remains susceptible to CTX lysogeny.

**Spatiotemporal and evolutionary dynamics of *Vibrio cholerae* in Cox's Bazar**

Isolates belonging to multiple Clade 5 subclades circulated concurrently in Cox's Bazar with genomes diverging by as few as zero SNPs (mean 13.8 SNPs) from isolates found sympatrically in the refugee camps and the adjacent unvaccinated host BGDN communities (Fig. 3a & b). There is clear evidence of sharing of almost identical isolates (minimum of 0 and 2 SNPs; mean 6.2 and 14.9 SNPs in subclade 3.9 and 5.20, respectively) between the FDMN and BGDN subpopulations within the refugee camps (Fig. 3c).

Looking more broadly across Bangladesh and the Bay of Bengal, all three Clade 3 and 5 subclades seen in Cox's Bazar were seen contemporaneously in Dhaka (Figs. 2b, 3d, Supplementary Fig. S4). This was more strongly observed in isolates taken from the BGDN population rather than any of the patients living in the refugee camps, with SNP distances between HP_BGDN and Dhaka isolates being comparable to within HP_BGDN distances (15.3 and 12.8 mean SNPs,

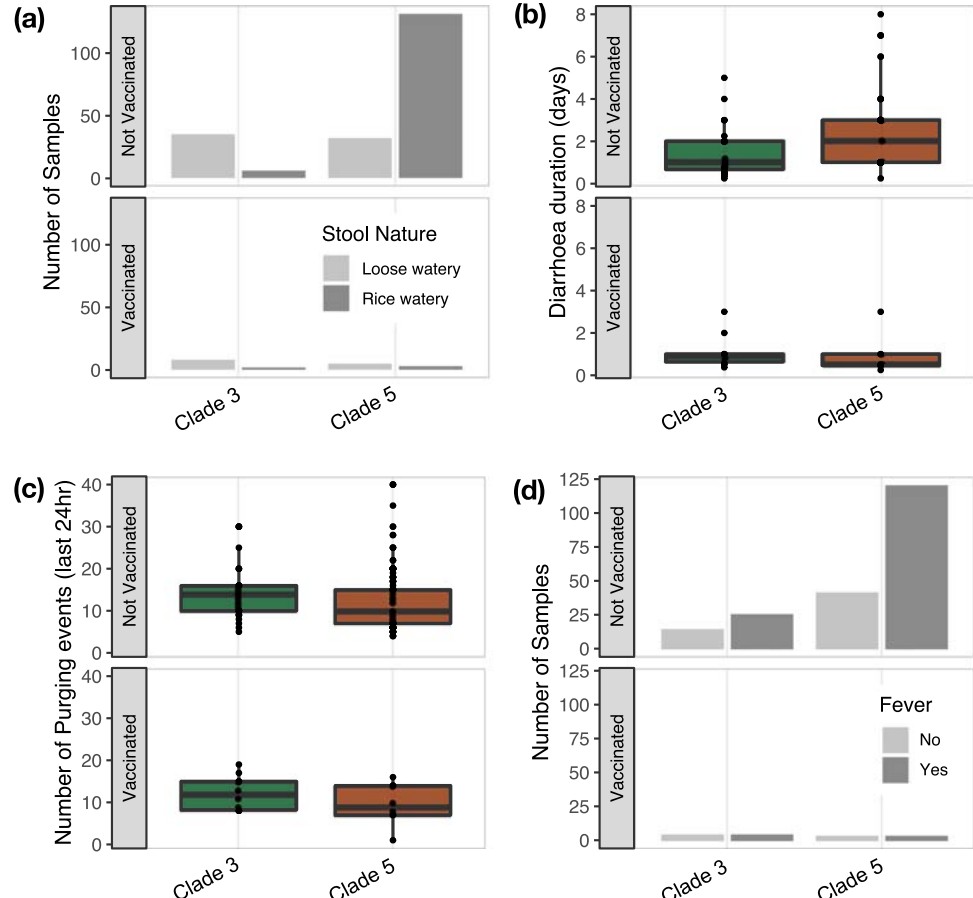

**Fig. 4 | Epidemiology of *V. cholerae* in Cox's Bazar.** Distribution of samples by stool nature (**a**), mean duration of diarrhoea (**b**), mean number of purging events (**c**), and distribution of samples by presence of fever (**d**) across vaccination status and phylogenetic clade. For (**b**) and (**c**), the box bounds represent the interquartile range (25th to 75th percentile), the midline represents the median, and the whiskers depict the minima and maxima, with individual values and outliers shown by the points.

respectively). This contrasted with the lack of subclade sharing between Cox's Bazar and Kolkata. Further, SNP distances between Dhaka and Cox's Bazar genomes were comparable to those observed for subpopulations within Cox's Bazar described above: for example, genomes from Dhaka and Cox's Bazar were separated by a mean of 8.6 SNPs in subclade 3.9.

To further explore the evolutionary relationships between these isolates, we calculated the SNP accumulation rates of parental clades, Clade 5 and Clade 3, to be 3.949 and 3.027 SNPs per genome per year, respectively ($R^2$ values of 0.427 and 0.804, respectively; Supplementary Fig. S5). This dated the most recent common ancestors (MRCAs) of Clade 5 and Clade 3 to 1993 (1991–1995; 95% confidence interval) and 1988 (1987–1989; 95% confidence interval), respectively. We then refined this analysis by focusing on the Cox's Bazar subclades: subclade 3.9 had an estimated evolutionary rate of 3.601 SNPs per genome per year, slighter faster than 2.709 for subclade 5.19 and 2.734 for subclade 5.20. This placed the MRCAs of subclade 5.19, subclade 5.20 and subclade 3.9 to be in ~2007, ~2009 and ~2016, respectively (Fig. 3d, Supplementary Fig. S5), consistent with the earliest genomes in each clade being isolated in 2009, 2011 and 2015, respectively. Based on the phylogeny and ancestral reconstruction of Clade 5 subtree, isolates occupying basal positions in the tree were more commonly isolated in Dhaka than in Cox's Bazar (Fig. 3a–c), suggesting that these lineages may have originated there. Sample size was too low for subclade 3.9 to conduct a similar analysis.

## Contrasting epidemiology of *Vibrio cholerae* clades in Cox's Bazar

Case notes classified AWD as either "rice watery stool" (RWS; watery stool, flecked with mucous, often pale to white in colour) or "loose watery stool" (LWS), with RWS being more frequent overall than LWS ($n = 143$ and $n = 80$, respectively; Supplementary Data S1). When considering the vaccine status, vaccinated patients had a statistically significantly lower incidence of RWS than LWS ($n = 13$ and 5; $p = 5.004 \times 10^{-4}$, Fishers Exact test), and diarrhoea lasted significantly longer in unvaccinated patients (up to eight days; average 1.973 days), compared to the vaccinated patients (only 1.009 days on average; $p = 2.882 \times 10^{-5}$, Mann–Whitney test). Only three of the vaccinated patients had severe dehydration, a hallmark of cholera, while 15 had only "some" dehydration ($p = 0.231$, Fishers Exact Test). There was no significant difference in the number of purging events (i.e. number of times passing stool) or reports of fever in vaccinated vs. unvaccinated patients ($p$-values of 0.336 and 0.163, respectively, Anova and Fishers Exact Test).

Stool nature also appeared to be associated with phylogenetic clade: LWS was more frequent than RWS in Clade 3 ($n = 43$ and 8, respectively; Fig. 4), while the converse was true for Clade 5 ($n = 37$ and 134, respectively), and this difference was significant (odds ratio of 19.14; $p = 5.217 \times 10^{-16}$, Fishers Exact Test; Fig. 4). Similarly, Clade 5 isolates were associated with a longer duration of diarrhea than those from Clade 3 (mean 2.044 days compared to 1.332 days, $p = 7.636 \times 10^{-5}$, Welch Two Sample t-test; Fig. 4). Although diarrhoea duration was longer in Clade 5, on average the number of purging

events was significantly higher in Clade 3 than Clade 5 (mean 14.22 events vs. 12.04 events; $p = 0.002$, Mann–Whitney test). Dehydration was recorded as severe in 34.5% of Clade 5 cases ($n = 59$), compared to only 11.7% of Clade 3 ($n = 6$) ($p = 7.25 \times 10^{-5}$, Fisher's exact test). Population sampled had a strong effect on both stool nature and diarrhoea duration: RWS was reported more frequently in the host population than the refugee camps (88.67% vs 1.370%; odds ratio of 0.021; $p = 2.2 \times 10^{-16}$, Fishers Exact Test), and diarrhoea lasted longer in the host population (2.095 vs 1.442 days; $p = 1.333 \times 10^{-8}$, Mann-Whitney test). However, samples from the two populations were not evenly distributed across the two clades, so we tested for covariates using logistic regression. In the best fit general linear model (AIC = 160.98), RWS was positively associated with the RC population ($p = 2.41 \times 10^{-7}$), Clade 3 ($p = 0.2363$), and not being vaccinated ($p = 0.1281$).

## Genetic microvariation within *V. cholerae* clades present in Cox's Bazar

To understand if genetic variation between clades or subclades may contribute to the differences in disease presentation or duration, we compared the pan genome of the Clade 5 and Clade 3 isolates. While no single variable trait defined either lineage, we did find evidence of within-lineage microvariation (Table 2), including frequent switching from serotype Inaba to Ogawa in isolates taken from BGDNs between July 2014 and October 2019 (Supplementary Fig. S6). This process appears to be mostly stochastic, although we observed a general trend toward Clade 5 being predominantly Inaba from 2016 while Clade 3 mainly contained Ogawa isolates.

The *ctxB1* allele of the gene coding the cholera toxin B-subunit was present in all Clade 5 isolate genomes, including our newly sequenced genomes (Fig. 2c, Supplementary Fig. S7). *ctxB1* was also characteristic of early strains of Clade 3 (up to 2005); however, following its emergence in 2006[32], *ctxB7* was also present in Clade 3, and after a period of co-circulation of the two genotypes, *ctxB7* completely replaced *ctxB1*. Hence, all our Clade 3 isolates from Cox's Bazar carried the *ctxB7* allele (Supplementary Fig. S7).

Conversely, unlike Clade 3, Clade 5 isolates showed heterogeneity in the gene complement of the *Vibrio* seventh pandemic island II (VSP-II). VSP-II has been previously classified into types B-D or 1-5[23,33]. All subclade 5.19 samples from our study ($n = 14$) carried VSP-II type 3/type C, which lacks orthologues of VC_00498 to VC_00502. This variant also dominated subclades 5.18, 5.22 and 5.23. We then observed two variants of VSP-II in our subclade 5.20 genomes: the majority of isolates ($n = 155$) carried type 4/type D (lacking orthologues of VC_0493 to VC_0497 and VC_00503, in addition to those lost from type 3), while

the remaining two carried type 3/type C (Supplementary Fig. S7). All 51 Clade 3 isolate genomes possessed VSP-II type 1 (lacking orthologues of VC_00498 to VC_00513).

Meanwhile, an SXT ICE most similar to ICE*Vch*Ind5/ICE*Vch*Ban5/RND6878/ICE[GEN] (accession number MK165650) – the dominant ICE in 7PET pandemic Wave 3 – was carried by both Clade 5 (32.8% of genomes) and Clade 3 (68.8% of genomes) (Supplementary Fig. S7). The lower prevalence in Clade 5 is due in part to its replacement by IDH_1986/ICE[TET] (accession number MK165649)[34] on a branch leading to a clade including part of subclade 5.22, and whole of subclades 5.19 and 5.20, which emerge from the former (Supplementary Fig. S7). All 14 subclade 5.19 isolates and 94.3% (148/157) of subclade 5.20 isolates possess the island IDH_1986/ICE[TET], in which a tetracycline resistance (*tetAR*) gene cassette replaced the florphenicol resistance (*floR*) gene cassette present in ICE*Vch*Ind5/RND6878/ICE[GEN] (Table 2, Supplementary Fig. S7). All 162 novel genomes from our study that carried ICE[TET] encoded *tetR*, while all 51 genomes carrying ICE[GEN] encoded *floR*. We were able to combine genotypic and phenotypic AMR data for a subset of the RC samples. This showed that of the sixteen isolates tested against tetracycline (TET), all seven samples showing phenotypic TET resistance encoded the *tetR* gene. Intriguingly, a further three TET-susceptible isolates encoded *tetR* on the ICE[TET]. All 57 isolates tested against ciprofloxacin were sensitive to it.

Finally, we detected the phage-inducible chromosomal island-like element PLE1 (KC152960.1; 95% coverage) in 92 isolates from Cox's Bazar (Supplementary Fig. S7). This element exhibits lineage-specificity, detected only in subclades 5.19 (10/14) and 5.20 (82/157). Outside our study, strains carrying this element emerged in 2010 and were detected sporadically in Bangladesh, Pakistan, and Iraq, and have not been detected since 2017[5,8,25,27]. In addition to this temporal signature, in Cox's Bazar, PLE1 exhibited a seasonal signal, being detected throughout the rainy season, and was particularly prevalent from June 2016 to January 2017.

## Discussion

We set out to determine if the *V. cholerae* responsible for the cases of cholera in Cox's Bazar around the time of the mass influx of Rohingya refugees belonged to the epidemic lineage known as seventh pandemic El Tor (7PET), responsible for almost all cases of epidemic cholera worldwide since the current pandemic began in 1961. Our data showed *V. cholerae* present in Cox's Bazar, including in the refugee camps, belonged to two distinct clades of epidemic Wave 3 of 7PET, denoted here as Clade 3 and Clade 5. Clade 3 and Clade 5 displayed contrasting spatiotemporal dynamics whereby Clade 5 has not been seen outside of South and Southeast Asia, whilst Clade 3 emerged in the Bay of Bengal and has spread globally (including introductions into Latin America, Africa, the Caribbean and the Middle East).

Within Bangladesh, our phylodynamic analysis strongly suggested that short-range local transmission is common in Cox's Bazar – including within refugee camps, as well as between the adjacent host population and the refugee camps – and between Cox's Bazar and Dhaka. However, our data shows that the overall number of lineages seen in Cox's Bazar was substantially lower than the number observed in Dhaka at the time of analysis, with ancestral reconstruction suggesting that the MRCA of Clade 5 originated in Dhaka. So whilst Dhaka appears to be part of a global transmission hub, Cox's Bazar may represent a transmission sink for the movement of isolates from Dhaka. This adds to a growing body of evidence that the Bay of Bengal and the Ganges Basin act as a diverse source of *V. cholerae* O1 capable of seeding epidemics worldwide, through cascades of short-range transmission or rarer, long-range transmission events[4,23–25,27].

It is important to note that because we cannot infer directionality between transmission pairs, one limitation of our study is the limited information on the burden of cholera in Myanmar. Whilst there were no known active cholera outbreaks in the Rakhine state at the time of

**Table 2 | Comparison of the major genetic differences between the two lineages detected in this study and correspondence with previous nomenclatures**

|  | Clade 5 | Clade 3 |
|---|---|---|
| **Major lineages** | none | T11, T12, T13, LAT3 |
| **Minor lineages** | Global 3.B, Asian lineage 1, Dhaka Clade B, Dhaka B5, Dhaka B6, PSC-1 | Global 3.C, Asian lineage 2, Dhaka B1, Dhaka Clade A, PSC-2, Haitian variant |
| **Serotype** | Predominantly Inaba | Predominantly Ogawa |
| ***ctxB* type** | *ctxB1* | Predominantly *ctxB7*[a] |
| **VSP-II type** | VSP-II type 2,3,4 | VSP-II type 1/C |
| **ICE-SXT** | SXT_ICE[TET]/IDH_1986[b] and SXT_ICE[GEN] RND6867[c] | SXT_ICE[GEN]/RND6867 |
| **PLE1** | PLE1[d] | none |

[a]Subclade 3.10 carries only ctxB1 and 3.12 carries predominantly ctxB1.
[b]Subclades 5.18, 5.19, 5.20, 5.22.
[c]Subclades 5.21, 5.22.
[d]Subclades 5.19, 5.20, 5.23.

the population displacement, the South Asian region is endemic for cholera[10,35], and hence these strains may have also originated from Myanmar. Notably, Myanmar experienced higher than normal rates of diarrheal disease in 2012 and 2013, and 23% and 14% of tested severe diarrheal cases were determined to be *V. cholerae* O1[36]. Molecular analysis showed that 34 isolates carried the classical *ctxB* type (*ctxB*1), suggesting the strains present in 2012 and 2013 may have belonged to Clade 5, which would fit with the dominance of this clade in South Asia. This underlines the urgent need for much deeper and routine genomic surveillance in Bangladesh and neighbouring countries, especially as human mobility due to conflict and climate change increases.

Importantly, we observed that Clade 3 and Clade 5 cause cholera disease with slightly different presentations and risk profiles. The Clade 5 strains were associated with the trademark choleraic rice watery stool, with a longer duration of diarrhoea and higher incidence of severe dehydration than Clade 3 strains, which more commonly present as loose watery stool with higher numbers of purging events over a shorter duration, with limited dehydration. Whilst such comparisons will need to be repeated with larger datasets and in other settings, this may suggest slightly different severity and modes of transmission of the pathogens belonging to these two clades. Differences in clinical manifestation between different sublineages of 7PET has not previously been explored.

In an attempt to understand differences in disease presentation, we linked clinical outcome to genetic profile. We observed that Clade 3 strains, like Clade 5, encoded *ctxB*1 (the classical type) until 2010, at which point it was replaced by *ctxB*7, explained by the presence of a point mutation at nucleotide position 59 (C to A; resulting in a histidine to asparagine amino acid change)[32,37]. *ctxB*7 has a structural alteration in the CTB signal peptide resulting in higher cholera toxin production[32]. This may explain the higher frequency of purging events and/or prevalence of LWS over RWS observed in patients infected with Clade 3 strains. Speculating further, it is possible that the increased frequency of purging associated with Clade 3 may result in increased rates of transmission and could therefore contribute to its global distribution, compared to Clade 5. Notably, the number of purging events was also lower in vaccinated patients than unvaccinated ones. Vaccination was also seen to decrease the duration of diarrhoea – from on average 1.97 days to 1.01 days – in patients in our study. This underlines the importance of vaccination in decreasing bacterial shedding, and by extension, in decreasing transmission rates and lowering epidemic risk in at-risk populations.

In addition to strain differences, our data also show that different groups living within Cox's Bazar might be at different levels of risk of more severe disease. For example, we observed a higher prevalence of RWS in the adjacent host population than the FDMNs, and a higher proportion of FDMNs had limited or severe dehydration than the BGDNs, both of which agree with results from a recent epidemiological study on the same population[11]. Faruque *et al.* also suggested that such differences might indicate different access to health care and/or sanitation infrastructure: BGDNs from the RC-adjacent communities sought care sooner than the FDMNs, and Oral Rehydration Solution (ORS) use was lower for FDMNs at 61.5% compared to 71.6% for the host population. This mirrored findings from our study, where out of the 15 patients recorded to have no dehydration, 14 were from the host BGDN population, further supporting the notion that this subpopulation has easier access to ORS than residents of the refugee camps.

Here we showed that the isolates causing sporadic disease in the Rohingya and Bangladesh host population belonged to the high-risk 7PET *V. cholerae* clone still causing large-scale cholera epidemics globally. Given the risk posed by this lineage, it's difficult not to conclude that the pre-emptive mass vaccination campaign played an important role in preventing an epidemic within a highly vulnerable community (with high population density, poor sanitation, malnourishment). Many factors differentiated the response in Cox's Bazar from comparable humanitarian crises elsewhere, where large-scale cholera epidemics linked to the same 7PET sublineage present in Cox's Bazaar did break out. For example, following the Haitian earthquake in 2010, a reactive OCV campaign was not established until eighteen months after the first cholera case where overall there were over 820,000 and -10,000 recorded cases and deaths respectively[30,31]. Similarly, the response in Yemen largely concentrated on case management rather than outbreak prevention[15], with case numbers surpassing 2.5 million by April 2022.

OCVs underpin the WHO's global strategy for both cholera outbreak prevention and response in endemic and epidemic settings. This strategy is threatened by increasing demand set against significant supply shortages, placing immense pressure on the global stockpile, not least because production of Shanchol™ will cease at the end of 2023. Although OCV coverage was very high in this campaign – 92% to 97% depending on dose and time lived in Cox's Bazar[38] – vaccine hesitancy has arisen as a challenge to be addressed in such populations[39]. Hence, education and engagement efforts within displaced communities and neighbouring host populations are also crucial to ensure vaccine campaigns are as effective as possible.

Given the clear patterns of local, regional and global transmission of highly related and readily transmissible sublineages of 7PET, the importance of reliable epidemiological and genomic surveillance systems cannot be understated for their role in targeting vaccine provision, especially given the current imbalance between OCV demand and supply and resultant inability to achieve meaningful protection in at-risk populations. Due to this, decisions regarding target populations are being made on the basis of the shortage, rather than taking a rational data-driven, best-practice approach informed by genomic surveillance. Such systems are the only methods to precisely track the spread of cholera in endemic and epidemic-risk settings and to assess how the vaccine is affecting the *V. cholerae* population as well as host immunity. Only by strengthening genomic epidemiological surveillance in at-risk populations can we maximise the impact of the dwindling vaccine stockpile.

## Methods

### Study sites, diarrhoeal disease sentinel surveillance, cholera case definition and ethics approval

Samples utilised in this study were obtained through nationwide diarrhoeal disease sentinel surveillance described in refs. 1 and [12]. Briefly, in May 2014, the icddr,b in conjunction with the IEDCR started diarrheal disease surveillance in 10 hospitals, one of which was based in Cox's Bazar. This program was expanded to 22 surveillance sites in 2016 across 21 districts. Surveillance sites were selected based on AWD cases and government data[40]. We utilised culture-confirmed *V. cholerae* isolates obtained from the ongoing cholera surveillance.

*V. cholerae* was isolated from stool samples from Bangladeshi Nationals (HP_BGDN; $n = 150$ and RC_BGDN; $n = 42$) and FDMNs (RC_FDMN; $n = 30$) residing in seven districts throughout Cox's Bazar between July 2015 and November 2019, with a break from May to June 2016 due to a funding gap. Patients were recruited if they met the AWD case definition and hence were suspected cholera case, defined by the WHO as:

- "Diarrhea (age <2 months): Changed stool habit from the usual pattern in terms of frequency (more than the usual number of purging) or nature of stool (more water than fecal matter);
- "Diarrhea cases (age ≥2 months): Any patient attending hospital with 3 or more loose or liquid stools within 24 h or 3 loose/liquid stools or fewer causing dehydration in the last 24 h.

Four patients who met the case definition and and had no other severe comorbidity (eg, severe acute respiratory illness, acute cardiovascular symptoms, or severe acute neurological disorder) were enrolled by the physician from Saturday to Wednesday each week. Two

patients with diarrhea aged less than five years old and two patients aged five years or older were enrolled each day; if the target number of patients in a particular age group was not met, we overenrolled in the other group to meet the target of four patients. Informed consent was obtained from all adult participants, or from participants/legal guardians if children younger than 18 years old. The surveillance protocol was approved by the Research Review Committee and Ethical Review Committee of icddr,b. Upon receiving consent, the physician collected the patient's sociodemographic characteristics, diet and medical histories, sanitation and hygiene information; and a stool sample was collected for testing, as outlined in[12]. These characteristics are detailed in Supplementary Data S1.

### Culture and phenotypic characterisation of the isolates
Stool specimens were transported in Cary-Blair transport media within 15 days to the laboratories in Dhaka at the icddr,b where they were processed as described in[38]. Briefly, specimens were streaked onto taurocholate-tellurite gelatin agar (TTGA) and incubated overnight at 37 °C to identify *V. cholerae*; specimens were also inoculated in alkaline peptone water for enrichment and incubated for an additional 18–24 h prior to plating on TTGA. Suspected *V. cholerae* colonies were serotyped with monoclonal antibodies specific to *V. cholerae* O1 (Ogawa and Inaba serotypes) and O139 serogroups.

### DNA preparation, genome sequencing and quality control
Genomic DNA was extracted from 25 ml cultures of *V. cholerae* isolates grown overnight at 37 °C in LB media, using the Wizard Genomic DNA Kit (Promega, Madison, WI, USA) according to manufacturer's instructions. Genomic DNA integrity and purity was confirmed by agarose gel electrophoresis and NanoDrop™ 2000 Spectrophotometer (Thermo Fisher Scientific, USA), respectively. Whole genome sequencing was carried out on the Illumina Hiseq2500 platform (Illumina, San Diego, CA, USA) at the Wellcome Sanger Institute (WSI), generating 150 bp paired-end reads. High quality read sets were confirmed using FastQC (http://www.bioinformatics.babraham.ac.uk) and Kraken2 (database complete as at 21/05/2015)[41].

### De novo genome assembly, annotation and pangenome analysis
Sequences were assembled de novo from short reads by the WSI Pathogens assembly pipeline using Velvet v1.2.10[42] with contigs annotated using Prokka v1.5[43] using automated pipelines (available under an open source license, GNU GPL 3 at https://github.com/sanger-pathogens/vr-codebase and https://github.com/sanger-pathogens/assembly_improvement). Annotated assemblies were used as input for pangenome analysis using Roary v3.13.0[44], without splitting paralogs (-s). For the species-level analysis, we used a core gene threshold of 99% to produce a core gene alignment.

### Phylogenetic, evolutionary and spatiotemporal analysis
Together with a collection of published genomes that represent *V. cholerae* diversity (Supplementary Data S2), we first constructed a core gene phylogenetic tree of the species. The pangenome analysis tool Roary was run on a total of 567 genomes, as described above, and we utilised the resulting core gene alignment, from which an alignment containing only the polymorphic positions was constructed using SNP-sites v2.5.1[45]. This was used as input for IQ-TREE v1.6.10[46,47] for maximum likelihood phylogenetic tree inference.

For the 7PET phylogenetic tree, we utilised a larger genome collection (Supplementary Data S1) and constructed a reference-based pseudogenome alignment. Briefly, reads were mapped against *V. cholerae* N16961 (accession numbers LT907989 and LT907990) using SMALT v0.7.4 (https://sourceforge.net/projects/smalt/), with the maximum insert size *i* set as 3 times the mean fragment size of the sequencing library. PCR duplicate reads were identified and flagged in the resulting BAM file using Picard v1.92 (https://broadinstitute.

github.io/picard/). Variants were detected using samtools mpileup v0.1.19[48] with parameters "-d 1000 -DSugBf", and a BCF file containing these variant sites was created using bcftools v0.1.19 (http://samtools.github.io/bcftools/), which was filtered to exclude low quality sites (quality <50, map_quality <3, af1 < 0.95, ratio <0.75, depth <4, depth_strand <2). The resulting pseudogenome alignment hence contained SNPs relative to the reference, and Ns at low certainty sites and deletions relative to the reference. The pseudogenome alignment was used as input for IQ-TREE for maximum likelihood phylogenetic tree construction using the TVM + F + ASC + R2 model (determined by first using the option '-m MF').

Trees were visualised and annotated in Phandango v 1.3.0[49] and iTOL v5[50] and v6[51] (available at https://jameshadfield.github.io/phandango/#/ and https://itol.embl.de/ respectively).

To delineate clades and subclades of the *V. cholerae* 7PET phylogenetic structure, we performed a hierarchical Bayesian Analysis of Population Structure (BAPS) as implemented in fastBAPS[52], using the "baps" prior, conditioned on the phylogeny outlined above. This was manually curated to better align BAPS clades with the tree structure.

We tested for a temporal signal in our dataset and subclades subsets using TempEst v1.5.3[53] and estimated the tMRCAs and evolutionary rates by calculating the linear regression (implemented in R) between the root-to-tip distances and year of isolation of each sample. Root to tip distances were plotted against year of isolation for each sample using the ggtree, ape, dplyr and ggplot2 packages in R[54–57]. The same packages were also used to plot the year of isolation against the subclade trees.

Longitudinal plots of BAPS PGs/clades over time were created in R, using the dplyr and ggplot2 packages. Spatial plots of sample counts were created using ggmap[58], over maptiles downloaded from Stamen Design (www.maps.stamen.com/toner), under a Creative Commons Attribution (CC BY 3.0) license (https://creativecommons.org/licenses/by/3.0/); Data by OpenStreetMap, under ODbL. We used either direct GPS co-ordinates obtained in this study, or latitude and longitude data for cities.

### Comparative genome analysis and characterisation of genes of interest
For genotypic AMR characterisation, the read sets for the 7PET dataset (*n* = 2606) were screened against the Comprehensive Antibiotic Resistance Database (CARD) using ARIBA[59]. We applied the same approach to identify the *ctxB* genotype using a custom database (using reference sequences described in Supplementary Data S3). We conducted BLAST analysis against a database containing published ICE sequences (Supplementary Data S3) to identify contigs resembling ICEs, using identity and coverage thresholds of 90 and 95%, respectively. Differential presence of genes encoded on VSPs was identified via pangenome analysis, obtained by running Roary as described above, on the 7PET dataset.

### Statistical analysis of epidemiological patterns
We analysed the statistical relationships between a number of variables using R base packages as follows[60]. For comparison of count data (number of samples) against binary/categorial variables (vaccination status, lineage or symptoms with only two categories e.g. stool nature, fever), we used Fisher's exact test. For comparison of continuous data (eg. duration of diarrhoea and number of purging events) against binary/categorical variables (vaccination status or lineage), we used a one-way ANOVA. We also tested for confounders of these relationships using generalised linear models in base R.

### Reporting summary
Further information on research design is available in the Nature Portfolio Reporting Summary linked to this article.

## Data availability

The read data generated in this study have been deposited in the ENA database under accession codes ERS3328687 to ERS4218290 (for example, https://www.ebi.ac.uk/ena/browser/view/ERS3328750), as provided in Supplementary data file S1. The metadata for this study is also provided in Supplementary data file S1. The sequence accession codes and metadata for previously published genomes is provided in Supplementary data file S1 and S2. The Comprehensive Antibiotic Resistance Database (CARD) database is available at https://card.mcmaster.ca/.

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

## Acknowledgements

We thank Matthew Dorman and Sally Kay for assistance with sample receipt and coordination; Matthew Dorman for help with curation of the genome collection and helpful discussion; Mat Beale, Anne Bishop for helpful discussion; Gal Horesh for help with plotting in R; and the Sanger Pipelines and Sanger Pathogen Informatics teams for support. This research was funded by the United Nations Children's Fund (UNICEF) (BGD/PCA201840/PD2019319). The authors thank the Bill and Melinda Gates Foundation, Food and Agriculture Organization, and the Government of the Kingdom of the Netherlands for their support of our research efforts. The icddr,b is also grateful to the Governments of Bangladesh, Canada, Sweden, and the United Kingdom for providing core/unrestricted support. Sequencing was supported by the Wellcome Sanger Institute. A.T.B., F.L and N.R.T. were supported by Wellcome funding to the Sanger Institute (#206194 and 108413/A/15/D). This research was funded in part by the Wellcome Trust (Grant numbers 206194 and 108413/A/15/D). For the purpose of open access, the authors have applied a CC-BY public copyright licence to any Author-Accepted Manuscript version arising from this submission.

## Author contributions

A.T.B., M.H.A., N.R.T. and F.Q. conceived the work and designed the study, M.K.A., A.I.K., M.T.I., N.A.T. were involved in the acquisition of samples and data, A.T.B., M.H.A., A.I.K. and F.L. analysed and interpreted the data, A.T.B., M.H.A., A.I.K. and F.L. drafted the work and N.R.T. and F.Q. substantively revised it. All authors have approved the submitted version and have agreed to be accountable for their contributions along with the accuracy and integrity of the work.

## Competing interests

The authors declare no competing interests.
