## [Peer Review File · Nature Communications]

Genomic epidemiology of *Vibrio cholerae* during a mass vaccination campaign of displaced communities in BangladeshREVIEWER COMMENTS

Reviewer #1 (Remarks to the Author):

Thank you for your important work and significant paper. It is well-written, compelling, and increasingly relevant as forced displacement accelerates. A few minor suggestions:

1. Discussion, Page 17, Lines 401-406. The authors state clearly the benefits of vaccination - decreased number of purging events, decreased duration of diarrhoea, decreased transmission rates. While noted briefly in the paper elsewhere, it would seem critically important to highlight low vaccination rates in forcibly displaced populations. It would also be beneficial to the reader to understand some of the reasons behind low vaccination rates (e.g., fear of authorities, strained relationships between host country population and displaced population, traumatic history of medical treatment of vulnerable refugees and asylum seekers); if possible, the reasons for low vaccination might be mentioned in the Introduction as well (Page 3, Line 60). I think it would also be important to note that while antibiotics (which the authors mention are more readily available than OCV, and which decrease the volume and duration of diarrhoea and bacterial shedding), are not always available in displaced or humanitarian emergency settings, and are often nearly impossible to find. This lack of access to antibiotics brings even more importance to the author's findings.

2. Discussion, Page 18, Lines 416-428. Access to care (e.g., ORS, mentioned by the authors), is extremely limited in humanitarian settings. While this is the point of this paragraph, it would be strengthened by quantifying the availability of ORS in these settings, perhaps in line 426 (putting a number to your statement that BGDN has easier access to ORS than residents of the RCs; perhaps not specific to this region of the world if the data do not exist, but generally the availability of ORS in humanitarian settings). Also, perhaps the last sentence could be framed as 2 sentences along the lines of: "Our data, together with previous reports, show that it is imperative to provide equitable access to vaccines for all at-risk populations. Equally important may be providing access to WaSH regimes, education, and access to treatment such as ORS." Just a suggestion, something along those lines.

3. Discussion, Page 19, Lines 435-443. I think it is worth mentioning that the *V. cholerae* that was brought to Haiti was brought by U.N. Peacekeepers. This is relevant because often in humanitarian settings, one has a flood of foreigners trying to contribute - and there is often unintentional harm in addition to the potential good that is done (i.e., the social theory of the unintended consequences of purposive social action). This theory has direct relevance to the argument you are making.

4. Discussion, Page 19, Lines 450-453. When noting the effectiveness of the vaccine for populations at highest risk, I think it would be worthwhile to note again that forcibly displaced people are a particularly vulnerable subset of individuals. Along the lines of, "...and populations at highest risk, such as forcibly displaced individuals/people."

Reviewer #2 (Remarks to the Author):

Taylor-Brown and colleagues have performed a robust and high-quality whole genome analysis of isolates of *Vibrio cholerae* causing human disease in and around Cox's Bazar in Bangladesh across two communities. The manuscript shows two dominant lineages with epidemic-causing potential circulating in the population and comparative genomic analysis shows allelic replacement of key virulence genes over time. I have no major concerns about the soundness of the bioinformatic/genomic analyses and my comments are mainly around how the results are presented:

Line 43: Please explain why Kolkata was used as a point of reference.

Line 47: Please define "Rice watery stool".

Line 57: What was the original population of Cox's Bazar before the arrival of FDMNs? If this is known, please include the number so that the percentage of population increase can be inferred.

Line 100: Please define what the "host community" refers to here. Is this the same as the HP_BGDN population you refer to later in the results?

Line 134: Please define the use of the word "Sadar" here.

Line 135: Please define "upazila".

Line 137: Please define "union" in this context

I found Table S2 to be quite useful to understand the first results section and I think should be provided as a table in the main manuscript. The location-specific terminology (upazila, union) could also be defined in the footnote of this table to help the readers.

Figure 1a - please define "SS" in Burmapara SS.

I could not find any specific reference to results shown in Figure 1b in the text. The sublineages described in Figure 1b are only introduced in Figure 2a, so this figure should come after Figure 2a.

Line 157: Please state which 7PET reference genomes clustering with the study genomes, used to define the 7PET lineage here, the tip labels are not clearly visible in this figure.

Supplementary Figure S1:

- Please state how this the tree is rooted in the figure legend.
- The tip labels in the figure in the tree are not visible so should be removed.

Figure 2:

- Figure 2a: I think there is too much information in such a small figure. This figure should be enlarged to fit at least half a page.
- Tip labels should be removed as they are not clearly visible.
- The title for the colour key "BAPS L1 PG[^] (subclade)" should be BAPS L2...?. I think there is the same typo in Figure S2.
- The figure legend says PG 1.7 is shown but the colour key says PG 1.8?
- Why is PG 1.8/1.7 included in this figure? I don't believe it is referenced anywhere in the text.

Line 163. It might not be immediately clear to non-hierBAP users that PG 2.5.19 is a sublineage of PG 1.5. For clarity, I suggest the first digit indicating BAPS level is not used.. (so that: PG 5.19 and PG 5.20 are sublineages of PG 5). Not critical- only a suggestion.

Line 169: Similar to my comment on Line 43, Why are these 2 cities mentioned specifically? Are Dhaka and Kolkata the only 2 nearby locations with isolates? Please clarify.

Supplementary Figure S3: Some of the labels are not fully visible.

Line 280-: I found this section quite interesting! And a nice summary of the findings in Table 1. I thought it might have been better to have this presented before the epidemiology results section - but this is only a suggestion.

Reviewer #3 (Remarks to the Author):

"Understanding the nature of *Vibrio cholerae* circulating prior to and during a mass vaccination campaign in the Forcibly Displaced Myanmar National (FDMN) population hosted in Cox's Bazar, Bangladesh." by Taylor-Brown et al, is a solid and important account of the authors' work to understand the complex cholera dynamics in Bangladesh and how it relates to the current global

cholera pandemic. The paper describes the whole-genome sequencing of 223 *V. cholera* isolates collected from Bangladeshi as well as displaced Myanmar nationals connected to refugee camps in Cox's bazaar, Bangladesh, before and after an oral vaccination campaign. All isolates belonged to the current pandemic 7PET lineage, and the paper further explores the relation to Dhaka/Kolkata and international strains circulating.

The paper is well written and largely an interesting read, but is to some extent overly reliant on a host of abbreviations. It would profit from a list of abbreviations chapter in order to help readers avoid incessant scrolling up and down. Although not necessarily a problem, I also think the paper could benefit from having a clearer conclusion. As it stands, it's meandering around a lot of different results without ever pulling them together into a unified conclusion. In fact I personally found that some of the analyses that were hidden away in the supplementary material were the most interesting, in particular supplementary figures 3 and 4. The paper does discuss risks of a larger epidemic in the absence of a vaccination and control campaign, but this does not take center stage in the paper.

The methods are technically sound for the most part, but I have some concerns about the use of statistics and inconsistencies in reporting as detailed below.

- Line 250: A fisher's test for association between vaccination and LWS/RWS is presented, and $N=13$ and $N=5$ is stated as the group sizes for vaccinated LWS/RWS patients, respectively. I do not understand how the presented p-value of $1.09e-6$ can come out of the mentioned numbers. (The total LWS/RWS numbers are presented as 80/143 in the lines before). Either there are some numbers or assumptions hidden here or the test has not been performed correctly. In fact, when I use the numbers from the supplementary table for Stool.Nature (Loose watery/Rice watery) and Vaccination.Status (Vaccinated/Not vaccinated) I get the numbers [58,137,13,5]. While significant, the associated p-value would be $5.0e-4$, not $1e-6$.
- More egregiously, in the next few lines, the authors perform an anova for the association between diarrhea duration and vaccination. Disregarding that it is weird to use anova for a two-category test, if one plots these data, it is clear that most of the assumptions underlying an anova/F-test are violated. In particular, the data are not normally distributed, are measured on a small-level discrete scale, and the variances vary.
- In line 260, Fishers exact test is used again, this time relating to an odds ratio of RWS vs LWS in PG1.5. It is not clear what is the second variable here. (Fisher's test requires at least two categories)
- Lines 265-270 - The statistical test is missing from all presentation of p-values.
- Lines 269-270 - These proportions are interesting, but do not say much since vaccination was not randomized across different refugee camps and local groups.
- Line 275 - GLMs are used for the association of RWS to different populations as well as duration. There is a wild difference in RWS frequency. Presumably, vaccination was not added to this model because that data was only present in a small subset of the study population. For duration, the associated p-value is 0.998, which seems extremely strange given that the means in the different groups are 2.1 and 1.44. Do the authors really mean to say that if they did 1000 replicates of the study, they would expect to find a larger difference in mean duration in 998 experiments under the assumption that there is no group difference? In the preceding sentence, the duration difference is reported as significantly different. I would strongly encourage a closer examination of the results and assumptions of this statistical model.
- Figure 2 (and phylogenetic trees in supplementary): There are some very long branches in these trees that make me worry about recombination, contamination or some other artefactual signal playing a part. It would probably be better to exclude these few isolates. These isolates will also mess up the evolutionary model that one is trying to fit, making the remainder of the tree less reliable. Another detail here is that the tree is stated to be outgroup-rooted, but the root is, as far as I can tell, never reported.

Nitpicks:

- I was surprised to see the use of Velvet to assemble the sequence data. This isn't wrong in any way, but it is certainly a program that would be considered a bit outdated by many bioinformaticians.
- Line 545-546: BAPS was used to hierarchically divide the phylogenetic tree. The BAPS results

were then manually curated to better correspond to the tree. I appreciate what is done here, but I find that the BAPS algorithm is already a fairly "black box-y" way to group isolates in your tree that you might have done equally well manually, and when you have to alter group membership afterwards I fail to see the point of using it in the first place.

Line 572: Incorrect reference (team, 2018)

- Figure 1 (and to some extent figure 3) has washed out colors that are hard to tell apart.

Structures like the Bangladesh-Myanmar border are annotated in the map but with the same style as rivers and roads.

REVIEWER COMMENTS

Reviewer #1 (Remarks to the Author):

Thank you for your important work and significant paper. It is well-written, compelling, and increasingly relevant as forced displacement accelerates. A few minor suggestions:

Comment 1. Discussion, Page 17, Lines 401-406. The authors state clearly the benefits of vaccination - decreased number of purging events, decreased duration of diarrhoea, decreased transmission rates. While noted briefly in the paper elsewhere, it would seem critically important to highlight low vaccination rates in forcibly displaced populations. It would also be beneficial to the reader to understand some of the reasons behind low vaccination rates (e.g., fear of authorities, strained relationships between host country population and displaced population, traumatic history of medical treatment of vulnerable refugees and asylum seekers); if possible, the reasons for low vaccination might be mentioned in the Introduction as well (Page 3, Line 60). I think it would also be important to note that while antibiotics (which the authors mention are more readily available than OCV, and which decrease the volume and duration of diarrhoea and bacterial shedding), are not always available in displaced or humanitarian emergency settings, and are often nearly impossible to find. This lack of access to antibiotics brings even more importance to the author's findings.

Author response: Thank you for raising these points. We have added some statements regarding OCV and antibiotic use to the introduction and discussion, for example:

Added the sentence (intro) “This stockpile is limited and thus heavily controlled, with provisions reserved for epidemic response rather than in endemic settings.”

Revised (discussion): “Antibiotics, which are **in general** more readily available than OCV, have been shown to decrease volume and duration of diarrhoea and bacterial shedding by up to 50% (Lindenbaum, Greenough, & Islam, 1967). **While this seems promising, the appropriate antibiotics are not always available or correctly prescribed, further demonstrating the need for vaccines.** Resistance to commonly used antimicrobials such as azithromycin and ciprofloxacin is an emerging threat to cholera”.

Comment 2. Discussion, Page 18, Lines 416-428. Access to care (e.g., ORS, mentioned by the authors), is extremely limited in humanitarian settings. While this is the point of this paragraph, it would be strengthened by quantifying the availability of ORS in these settings, perhaps in line 426 (putting a number to your statement that BGDN has easier access to ORS than residents of the RCs; perhaps not specific to this region of the world if the data do not exist, but generally the availability of ORS in humanitarian settings). Also, perhaps the last sentence could be framed as 2 sentences along the lines of: "Our data, together with previous reports, show that it is imperative to provide equitable access to vaccines for all at-risk populations. Equally important may be providing access to WaSH regimes, education, and access to treatment such as ORS." Just a suggestion, something along those lines.

Author response: Data about access to care is limited, however, we have added some numbers from Faruque et. al (that was already cited) around ORS use in this setting (line 469-470): “Oral Rehydration Solution (ORS) use was lower for FDMNs **at 61.5% compared to 71.6% for the host population**”.

Comment 3. Discussion, Page 19, Lines 435-443. I think it is worth mentioning that the V. cholerae that was brought to Haiti was brought by U.N. Peacekeepers. This is relevant

because often in humanitarian settings, one has a flood of foreigners trying to contribute - and there is often unintentional harm in addition to the potential good that is done (i.e., the social theory of the unintended consequences of purposive social action). This theory has direct relevance to the argument you are making.

Author response: Whilst this point is important, we have not made this change because based on the comments from all reviewers we have streamlined the discussion and now this is no longer relevant to include.

Comment 4. Discussion, Page 19, Lines 450-453. When noting the effectiveness of the vaccine for populations at highest risk, I think it would be worthwhile to note again that forcibly displaced people are a particularly vulnerable subset of individuals. Along the lines of, '...and populations at highest risk, such as forcibly displaced individuals/people.'

Author response: We extended the sentence as suggested (line 474): "and populations at highest risk, **such as those that have been forcibly displaced.**"

Reviewer #2 (Remarks to the Author):

Taylor-Brown and colleagues have performed a robust and high-quality whole genome analysis of isolates of *Vibrio cholerae* causing human disease in and around Cox's Bazar in Bangladesh across two communities. The manuscript shows two dominant lineages with epidemic-causing potential circulating in the population and comparative genomic analysis shows allelic replacement of key virulence genes over time. I have no major concerns about the soundness of the bioinformatic/genomic analyses and my comments are mainly around how the results are presented:

Comment 1. Line 43: Please explain why Kolkata was used as a point of reference.

Author response: Dhaka and Kolkata are two key sites for the study of cholera in the Bay of Bengal, for which there is relevant and extensive genomic data and where there are known to be co-circulating lineages. Our genomes fell within the same clades as recently sequenced genomes from both cities and hence we felt it important to put these data in the broader context of what's known and published from both Dhaka and Kolkata.

Comment 2. Line 47: Please define "Rice watery stool".

Author response: The word limit of the abstract does not permit a more detailed explanation, however we have added some further details in the results (line 278-279): "Case notes classified AWD as either "rice watery stool" (RWS; **watery stool, flecked with mucous, often pale to white in colour**) or "loose watery stool" (LWS)."

Comment 3. Line 57: What was the original population of Cox's Bazar before the arrival of FDMNs? If this is known, please include the number so that the percentage of population increase can be inferred.

Author response: FDMNs have been seeking refuge in Cox's Bazar since the early nineties. In 2016 there were an estimated 444K host population and 860K Rohingya. we have added "...total population swelling to over a million people, **with FDMNs now outnumbering the host population by a factor of 2**" (line 82-83).

Comment 4. Line 100: Please define what the "host community" refers to here. Is this the same as the HP_BGDN population you refer to later in the results?

Author response: Thank you for pointing this out; we have changed this to “host population” to be consistent with the rest of the manuscript.

Comment 5. Line 134: Please define the use of the word "Sadar" here.

Author response: Cox's Bazar Sadar is an upazila within the Coxs Bazar district. I have clarified this by adding “...Cox's Bazar Sadar, Ukhia, and Teknaf upazilas (**administrative divisions within Cox's Bazar**; Figure 1)”.

Comment 6. Line 135: Please define "upazila".

Author response: Please see response to comment 6.

Comment 7. Line 137: Please define "union" in this context

Author response: We have added “...originated in Teknaf union (**small settlement**)...”

Comment 8. I found Table S2 to be quite useful to understand the first results section and I think should be provided as a table in the main manuscript. The location-specific terminology (upazila, union) could also be defined in the footnote of this table to help the readers.

Author response: Thank you – we have moved it into the main manuscript. We have addressed the location-specific terminology in the main text (see response to Comment 5 & 7 above), and have added the definitions to the footnotes for further clarity.

Comment 9. Figure 1a - please define "SS" in Burmapara SS.

Author response: We have added this to the figure legend.

Comment 10. I could not find any specific reference to results shown in Figure 1b in the text. The sublineages described in Figure 1b are only introduced in Figure 2a, so this figure should come after Figure 2a.

Author response: Thank you for pointing this out. This was a mistake. We have now changed the panels for figures 1 and 2 to better reflect the flow of the manuscript. (Please also see response to comment 13).

Comment 11. Line 157: Please state which 7PET reference genomes clustering with the study genomes, used to define the 7PET lineage here, the tip labels are not clearly visible in this figure.

Author response: We have removed the tip labels, and instead marked the 7PET genomes used for reference, including the commonly used reference genome, N16961 in grey, and updated this information in the figure legend. We also clarified this in the text: “All isolates sequenced in this study fell within the 7PET lineage, **clustering near *V. cholerae* reference genome N16961** (Supp Figure S1)”.

Comment 12. Supplementary Figure S1:

a- Please state how this the tree is rooted in the figure legend.

b- The tip labels in the figure in the tree are not visible so should be removed.

Author response: We have revised this tree and legend as per reviewer suggestions. The tree is midpoint rooted and the 7PET reference genomes (including N16961; listed in Supp Table S2 which was accidentally omitted upon submission) are highlighted more clearly.

Comment 13. Figure 2:

a- Figure 2a: I think there is too much information in such a small figure. This figure should be enlarged to fit at least half a page.

b- Tip labels should be removed as they are not clearly visible.

c- The title for the colour key "BAPS L1 PG[^] (subclade)" should be BAPS L2...?. I think there is the same typo in Figure S2.

d- The figure legend says PG 1.7 is shown but the colour key says PG 1.8?

e- Why is PG 1.8/1.7 included in this figure? I don't believe it is referenced anywhere in the text.

Author response: We have stripped some of the excess information from this figure as suggested to make it easier to digest, and revised the details in line with suggested changes to nomenclature and labelling.

Comment 14. Line 163. It might not be immediately clear to non-hierBAP users that PG 2.5.19 is a sublineage of PG 1.5. For clarity, I suggest the first digit indicating BAPS level is not used.. (so that: PG 5.19 and PG 5.20 are sublineages of PG 5). Not critical- only a suggestion.

Author response: We have revised the nomenclature as per reviewer comments, so that we now have, for example, clade 5 and subclade 5.19 and 5.20. We have also revised the text to better explain the hierarchical nature of the baps levels (line 154, 155).

Comment 15. Line 169: Similar to my comment on Line 43, Why are these 2 cities mentioned specifically? Are Dhaka and Kolkata the only 2 nearby locations with isolates? Please clarify.

Author response: Please see response to comment 1.

Comment 16. Supplementary Figure S3: Some of the labels are not fully visible.

Author response: We have revised all figures with clarified nomenclature so these are now visible.

Comment 17. Line 280-: I found this section quite interesting! And a nice summary of the findings in Table 1. I thought it might have been better to have this presented before the epidemiology results section - but this is only a suggestion.

Author response: Thank you! We moved these sections back and forth, and decided on this order as we could use the patterns of gene presence to explain the epidemiology we described in the preceding section.

Reviewer #3 (Remarks to the Author):

"Understanding the nature of *Vibrio cholerae* circulating prior to and during a mass vaccination campaign in the Forcibly Displaced Myanmar National (FDMN) population hosted in Cox's Bazar, Bangladesh." by Taylor-Brown et al, is a solid and important account of the authors' work to understand the complex cholera dynamics in Bangladesh and how it relates to the current global cholera pandemic. The paper describes the whole-genome sequencing of 223 *V. cholera* isolates collected from Bangladeshi as well as displaced Myanmar nationals connected to refugee camps in Cox's bazaar, Bangladesh, before and after an oral vaccination campaign. All isolates belonged to the current pandemic 7PET lineage, and the paper further explores the relation to Dhaka/Kolkata and international strains circulating.

Comment 1. The paper is well written and largely an interesting read, but is to some extent overly reliant on a host of abbreviations. It would profit from a list of abbreviations chapter in order to help readers avoid incessant scrolling up and down. Although not necessarily a problem, I also think the paper could benefit from having a clearer conclusion. As it stands, it's meandering around a lot of different results without ever pulling them together into a unified conclusion. In fact I personally found that some of the analyses that were hidden away in the supplementary material were the most interesting, in particular supplementary figures 3 and 4. The paper does discuss risks of a larger epidemic in the absence of a vaccination and control campaign, but this does not take center stage in the paper.

Author response: We appreciate this positive feedback! To address this point, we have been through the discussion to improve the focus. We have also made some changes to the main figures by including panels from the supplementary data to provide a better overall view of the main findings.

Comment 2. The methods are technically sound for the most part, but I have some concerns about the use of statistics and inconsistencies in reporting as detailed below.

a- Line 250: A fisher's test for association between vaccination and LWS/RWS is presented, and $N=13$ and $N=5$ is stated as the group sizes for vaccinated LWS/RWS patients, respectively. I do not understand how the presented p-value of $1.09e-6$ can come out of the mentioned numbers. (The total LWS/RWS numbers are presented as 80/143 in the lines before). Either there are some numbers or assumptions hidden here or the test has not been performed correctly. In fact, when I use the numbers from the supplementary table for Stool.Nature (Loose watery/Rice watery) and Vaccination.Status (Vaccinated/Not vaccinated) I get the numbers [58,137,13,5]. While significant, the associated p-value would be $5.0e-4$, not $1e-6$.

You are correct - this was an error - I had not excluded the "not known" vaccination category from the test. We have corrected this in the manuscript.

b- More egregiously, in the next few lines, the authors perform an anova for the association between diarrhea duration and vaccination. Disregarding that it is weird to use anova for a two-category test, if one plots these data, it is clear that most of the assumptions underlying an anova/F-test are violated. In particular, the data are not normally distributed, are measured on a small-level discrete scale, and the variances vary.

Thank you for pointing this out. We have now run a Mann-Whitney test as the data was not normally distributed, and have updated the values in the manuscript.

c- In line 260, Fishers exact test is used again, this time relating to an odds ratio of RWS vs LWS in PG1.5. It is not clear what is the second variable here. (Fisher's test requires at least two categories)

The two categories here are stool nature - LWS or RWS, within PG 1.5. We have re-written this section to clarify this.

d- Lines 265-270 - The statistical test is missing from all presentation of p-values.

Thank you for noting this - we have added these in.

e- Lines 269-270 - These proportions are interesting, but do not say much since vaccination was not randomized across different refugee camps and local groups.

Agreed. We have removed lines 267-271 to avoid this confusion.

f- Line 275 - GLMs are used for the association of RWS to different populations as well as duration. There is a wild difference in RWS frequency. Presumably, vaccination was not added to this model because that data was only present in a small subset of the study population. For duration, the associated p-value is 0.998, which seems extremely strange given that the means in the different groups are 2.1 and 1.44. Do the authors really mean to say that if they did 1000 replicates of the study, they would expect to find a larger difference in mean duration in 998 experiments under the assumption that there is no group difference? In the preceding sentence, the duration difference is reported as significantly different. I would strongly encourage a closer examination of the results and assumptions of this statistical model.

Author response: We have re-examined the models and revised this section. The difference in mean duration between the two populations is significantly different, and we have now accounted for the bias in distribution of population across the two clades.

Comment 3 - Figure 2 (and phylogenetic trees in supplementary): There are some very long branches in these trees that make me worry about recombination, contamination or some other artefactual signal playing a part. It would probably be better to exclude these few isolates. These isolates will also mess up the evolutionary model that one is trying to fit, making the remainder of the tree less reliable. Another detail here is that the tree is stated to be outgroup-rooted, but the root is, as far as I can tell, never reported.

Author response: The long branches have been previously reported, hence to be selective and remove them would be inappropriate for this manuscript because they fall within our criteria for inclusion in the tree.

Nitpicks:

Comment 4 - I was surprised to see the use of Velvet to assemble the sequence data. This isn't wrong in any way, but it is certainly a program that would be considered a bit outdated by many bioinformaticians.

Author response: These assemblies were constructed using the automated Sanger Pathogens Pipeline; I compared a subset of assemblies from SPAdes and they did not change or improve on the findings presented here.

Comment 5 - Line 545-546: BAPS was used to hierarchically divide the phylogenetic tree. The BAPS results were then manually curated to better correspond to the tree. I appreciate what is done here, but I find that the BAPS algorithm is already a fairly "black box-y" way to group isolates in your tree that you might have done equally well manually, and when you have to alter group membership afterwards I fail to see the point of using it in the first place.

Author response: We appreciate this concern, and acknowledge that no tool is perfect for this task, but BAPS places some rigour over and above visual assignment that is valuable here.

Comment 6 - Line 572: Incorrect reference (team, 2018)

Author response: This was an EndNote artefact and I have fixed the error.

Comment 7 - Figure 1 (and to some extent figure 3) has washed out colors that are hard to tell apart. Structures like the Bangladesh-Myanmar border are annotated in the map but with the same style as rivers and roads.

Author response: We apologise that some of the details are difficult to differentiate - these are artefacts of the way the map is rendered at the desired resolution and we unfortunately have no control over that.

REVIEWERS' COMMENTS

Reviewer #1 (Remarks to the Author):

Thank you for this important paper and the relevant changes. The revisions adequately satisfy the asks - thank you for your careful attention to the reviews.

Reviewer #2 (Remarks to the Author):

Thank you for responding to all my comments. I have no further comments.

Reviewer #3 (Remarks to the Author):

Thank you for addressing all my concerns. I believe the paper has improved significantly, and I have no further comments.

-Ola Brynildsrud

Reviewer #4 (Remarks to the Author):

Thank you for giving me the chance to read Taylor-Brown et al's paper characterising the epidemic risk of cholera strains in FDMN populations in Cox's Bazaar. It is well-written, engaging and extremely relevant given the increase in displaced populations worldwide.

I've been asked to give input on the study context. I only have one minor comment:

In the discussion section (from line 437-448), the authors comment on how a large proportion of those with no dehydration were from host country nationals, perhaps because they present earlier and use of ORS is more common in that population. To me this highlights the need for an effective OCV campaign for the FDMN, given it has been shown to lower the risk of dehydrating cholera. I wonder if a comment should be made about the challenges of vaccine hesitancy in this population? Studies have reflected vaccine hesitancy among FDMN, with resistance from religious leaders and gender barriers being some contributing factors, as well as a general mistrust of local health systems. These barriers are vital overcome to ensure adequate vaccine uptake.

REVIEWERS' COMMENTS

Reviewer #1

Thank you for this important paper and the relevant changes. The revisions adequately satisfy the asks - thank you for your careful attention to the reviews.

Reviewer #2

Thank you for responding to all my comments. I have no further comments.

Reviewer #3

Thank you for addressing all my concerns. I believe the paper has improved significantly, and I have no further comments.

Author response to reviewers 1-3: We appreciate the reviewers' positive feedback!

Reviewer #4

Thank you for giving me the chance to read Taylor-Brown et al's paper characterising the epidemic risk of cholera strains in FDMN populations in Cox's Bazaar. It is well-written, engaging and extremely relevant given the increase in displaced populations worldwide.

I've been asked to give input on the study context. I only have one minor comment:

In the discussion section (from line 437-448), the authors comment on how a large proportion of those with no dehydration were from host country nationals, perhaps because they present earlier and use of ORS is more common in that population. To me this highlights the need for

an effective OCV campaign for the FDMN, given it has been shown to lower the risk of dehydrating cholera. I wonder if a comment should be made about the challenges of vaccine hesitancy in this population? Studies have reflected vaccine hesitancy among FDMN, with resistance from religious leaders and gender barriers being some contributing factors, as well as a general mistrust of local health systems. These barriers are vital overcome to ensure adequate vaccine uptake.

Author response: Thank you for raising these points. We agree there are several behavioural barriers to overcome in order for vaccine campaigns to be as effective as possible. The OCV campaign did indeed target the FDMNs – only FDMNs were vaccinated, and in fact vaccine coverage was very high – 94 and 92% for the first and second dose, with 97% coverage of dose one in FDMNs that have lived in Cox’s Bazar for 4-6 months. We believe part of this success was due to the role of Majhis in educating and coordinating the vaccine effort. We have added the following sentences to the second last paragraph: **“Although OCV coverage was very high in this campaign – 92% to 97% depending on dose and time lived in Cox’s Bazar (Khan et al., 2019) – vaccine hesitancy has arisen is a challenge to be addressed in such populations (Jalloh et al., 2019). Hence, education and engagement efforts within displaced communities and neighbouring host populations are also crucial to ensure vaccine campaigns are as effective as possible.”**